# Therapeutic Phage Display-Derived Single-Domain Antibodies for Pandemic Preparedness

**DOI:** 10.3390/antib12010007

**Published:** 2023-01-14

**Authors:** Janet M. Daly, Theam Soon Lim, Kevin C. Gough

**Affiliations:** 1School of Veterinary Medicine and Science, University of Nottingham, Nottingham NG7 2RD, UK; 2Institute for Research in Molecular Medicine, Universiti Sains Malaysia, Penang 11800, Malaysia

**Keywords:** VHH, single-domain antibody, virus, therapeutic

## Abstract

Driven by necessity, the COVID-19 pandemic caused by SARS-CoV-2 has accelerated the development and implementation of new vaccine platforms and other viral therapeutics. Among these is the therapeutic use of antibodies including single-domain antibodies, in particular the camelid variable heavy-chain fragment (VHH). Such therapies can provide a critical interim intervention when vaccines have not yet been developed for an emerging virus. It is evident that an increasing number of different viruses are emerging and causing epidemics and pandemics with increasing frequency. It is therefore imperative that we capitalize on the experience and knowledge gained from combatting COVID-19 to be better prepared for the next pandemic.

## 1. Introduction

Single-domain antibodies (sdAb) are the smallest fragments of antibody that retain the full antigen-binding capacity of a conventional antibody. The majority of sdAbs are derived from camelid heavy-chain antibodies. In addition to conventional antibodies, members of the Camelidae family also produce heavy-chain-only antibodies containing a single variable domain instead of two variable domains (VH and VL) found in the antigen-binding fragment (Fab) of conventional immunoglobulin G antibodies (Figure 1). The first nanobody-based therapy (calpacizumab) was approved in the European Union in 2018 and in the USA in 2019 and for the treatment of acquired thrombotic thrombocytopenic purpura (aTTP), a rare blood-clotting disorder associated with an over-production of the von Willebrand factor. Calpacizumab is a bivalent, humanized variable domain antibody targeting the A1 domain of the von Willebrand factor, which inhibits interaction with the platelet glycoprotein Ib-IX-V receptor [1]. Viral targets for which VHH antibodies have been developed include SARS-CoV-2, the causative agent of COVID-19 [2], Middle East respiratory syndrome Coronavirus (MERS-CoV) [3], betacoronavirus [4] and influenza A virus [5].

## 2. Phage Display of VHH-Based Libraries for Pandemics

Like many other display-based recombinant antibodies, VHH are commonly derived from a library of antibody genes that are displayed on phage particles by panning against the target of interest. Different approaches to generating a VHH phage library include the immunization of an alpaca or llama and cloning of the VHH gene amplified by a PCR using reverse-transcribed RNA extracted from isolated lymphocytes into phagemid. Alternatively, VHH that bind the target protein of interest can be selected by panning against a ‘naïve’ or synthetic VHH library. Ye et al. [7] generated a naïve phage display library with a diversity of 7.5 × 10^10^ different VHH sequences using B cells from the spleen, bone marrow and blood of nearly a dozen non-immunized llamas and alpacas. In another approach, Tsoumpeli et al. [8] used a PCR to introduce diversity into the CDR3 of a camelid VHH scaffold known to function as an intrabody.

The possibility of using either naïve or immune libraries derived from closely related viruses in case of a pandemic for immediate antibody development is vital. This is essential, especially in cases where a rapid response is required. It is foreseeable that during the onset of a pandemic, a rapid response can be mobilized by first developing VHH antibodies using naïve libraries because it affords a shorter time frame from screening to antibody identification compared to traditional hybridoma technologies. The smaller size VHH allows for easy production using a prokaryotic host to meet worldwide demands, which is more difficult with mammalian cell expression systems. Although VHH antibodies isolated from naïve or synthetic libraries are generally lower in affinity, the need to undergo affinity maturation of just a single domain would be less complicated for engineering than a standard two-domain antibody. However, the potential cost implications involved in the removal of endotoxin contamination in prokaryotic expression systems could be a hindering factor for large-scale production of the VHH antibodies in the long run. Therefore, the application of alternative expression systems such as algae (spirulina) [9], *Nicotiana benthamiana* plants [10] or by in vivo delivery (mRNA therapy) could potentially reduce the cost of implementing VHH antibodies into society.

There are several publications showing the use of immunized camelid-based libraries for COVID-19 response [11,12,13]. Although antibodies generated by immune libraries generally exhibit higher affinities, good quality neutralizing antibodies have also been reported from naïve and synthetic VHH libraries [14]. Whether derived from naïve or immunized libraries, VHH will have to undergo humanization for therapeutic applications in line with regulations. Therefore, the development of VHH antibodies much like other antibody formats would require the design and planning of a suitable selection strategy depending on the planned downstream application of the antibody.

## 3. Advantages and Disadvantages of VHH

Until recently, most progress in the development of single-domain antibodies was in the field of cancer diagnostics and treatment (see review [8]). Characteristics of VHH that mean they have been avidly pursued in the cancer field are equally applicable to treatment of viral diseases including COVID-19. The VHH CDR3 are generally longer than conventional VH domains thus providing a larger epitope coverage and compensating for the loss of the VL domain repertoire [15]. The relatively small size of the VHH domains also allows them to bind occluded epitopes that conventional vaccines may not be able to elicit antibodies against [16]. This is critical as these epitopes are generally conserved with very low variability making them less subject to immunity-based selection pressures. An example is the binding of VHH to a highly conserved epitope in the receptor-binding domain (RBD) of the SARS-CoV-2 spike protein [17]. Their small size also means VHH have drug-like properties in terms of bioavailability [6], although they can be cleared too rapidly and need to be modified, for example by fusing with conventional Fc to improve their pharmacokinetic (PK) profile. It is also easier to select specific binders against conformational epitopes; as for other viruses, the majority of neutralizing antibodies against SARS-CoV-2 spike protein recognized conformational rather than linear epitopes [18]. Production of VHH is relatively easy to scale up making it ideal for immediate and large-scale production for global consumption. Their smaller size means expression in bacteria is easier; conventional antibodies require assembly of four polypeptide chains and extensive disulphide bond formation for functionality.

Recently, a paper reported the isolation of VHH antibodies against SARS-CoV-2 using an immune alpaca library with a high thermostability of up to 95 °C. Additionally, the reported VHH antibody also has an affinity in the picomolar range, which makes it very attractive. More importantly, the restricted use of just a single binding domain did not restrict the broadly neutralizing effects as it was able to withstand immune-escape mutations found in the alpha, beta, gamma, epsilon, iota, and delta/kappa lineages [13]. Critically, this suggests that the presence of only a single binding domain in VHH antibodies is sufficient to generate good neutralizing anti-infectives.

Availability of the DNA sequence encoding the phage-displayed antibody fragment means the selected antibody fragments can be rapidly further engineered, for example to achieve affinity maturation. Zupancic et al. [19] used an error-prone PCR and selection of clones with improved affinity of the SARS-CoV-2 spike protein, but also accidentally discovered that swapping the three CDRs between low-affinity clones also produced VHH with improved affinity. Schepens et al. [18] used a protein-modelling approach to identify a single amino-acid change that increased the affinity of VHH for the SARS-CoV-2 RBD. There are also straightforward approaches to ‘humanize’ camelid VHH domains [8,14], which is important for reducing immunogenicity to the scaffold [20]. With further engineering, two or three VHHs can be combined into a single polypeptide chain, without compromising folding or binding affinities [21,22], to generate neutralizing antibodies [23]. Engineering of the VHH domains to generate fusion molecules such as VHH-Fc and multi-domain VHH can improve their half-lives [24]. As they are more soluble and stable, VHH can be nebulized and administered using an inhaler to target the respiratory tract in the case of SARS-CoV-2 [25,26]. They can also be lyophilized and could be stockpiled in case of a future epidemic/pandemic.

A key feature of the COVID-19 pandemic has been the need to keep up with variants. It appears that VHH may have an advantage in this respect. So far, the only receptor-binding antibody that is cross-neutralizing is a VHH (VHH-72), which was produced by repeatedly immunizing a llama with SARS-CoV-1 and MERS-CoV spike proteins [4]. Multivalent and multi-specific molecules combining two or more antibodies might not only improve efficacy but also protect against resistance due to virus-escape mutants [27].

A potential Achilles’ heel that became apparent during a phase I clinical trial of a therapeutic VHH was that aggregation can result in (liver) toxicity and immune reactivity in patients with pre-existing anti-drug antibodies. This however can be addressed by developing in vitro assays to screen for aggregation-resistant VHH [8] and modifications to improve stability, for instance adding a negatively charged tail to the C-terminus can reduce aggregation [28].

## 4. Application of VHH in Pandemic Response

The recent COVID-19 pandemic has highlighted the shortcomings of the health response mechanism in many countries. Although a PCR-based diagnostic was the gold standard diagnostic recommended by the World Health Organization (WHO), it was evident that PCR-based diagnostics were not the solution for the rapid testing of a global population. Ultimately, when considering that the pandemic does not discriminate between developed and under-developed countries, cheaper and less complicated testing was necessary to ensure the greater population receives testing. The general characteristics of VHH antibodies discussed earlier highlights how VHH antibodies can be applied for the development of alternative diagnostic platforms. The most common and reliable approach that is cost-effective and easy to perform is the lateral flow assay.

VHH antibodies have been shown to work well in lateral flow assays and other immunoassays for the detection of infectious agents such as norovirus [29] and trypanosomes [30]. In the case of COVID-19, VHH antibodies have been applied in lateral flow [31] and even hemagglutination assays [32].

During pandemics, especially when dealing with new and unknown infections, there is a need to rapidly obtain a better understanding of the infection and research reagents like antibodies are critical to enable researchers to carry out faster studies. Hence, VHH antibodies can help fill that void especially when dealing with new infections where the reagent supply or availability is scarce. Antibodies developed recombinantly have the advantage of a faster turnaround time if compared to animal-immunization-based polyclonal antibodies and the even longer time required for hybridoma production due to the complexity of screening. Therefore, recombinant antibodies like VHH would provide an added advantage in this respect to ensure preparedness.

## 5. Prospects

In response to the emergence of COVID-19, the World Health Organization activated its global strategy and preparedness plan (the R&D Blueprint). This provides a roadmap for the rapid activation of efforts to develop effective tests, vaccines, and medicines against a list of priority diseases. The current R&D Blueprint list includes COVID-19; Crimean-Congo haemorrhagic fever; Ebola virus disease and Marburg virus disease; Lassa fever; MERS-CoV and SARS; Nipah and henipaviral diseases; Rift Valley fever; Zika; and “Disease X” (to allow for emergence of an unknown pathogen). With the recent emergence of mpox (previously called monkeypox) infections in new regions, there is also a possibility that novel VHH antibodies would be developed to establish rapid diagnostics and potentially neutralizing antibodies for therapy. Developing VHH against known viruses and identifying effective cross-reactive antibodies that could be used if related viruses emerge could be a key component of future pandemic preparedness.

## Figures and Tables

**Figure 1 antibodies-12-00007-f001:**
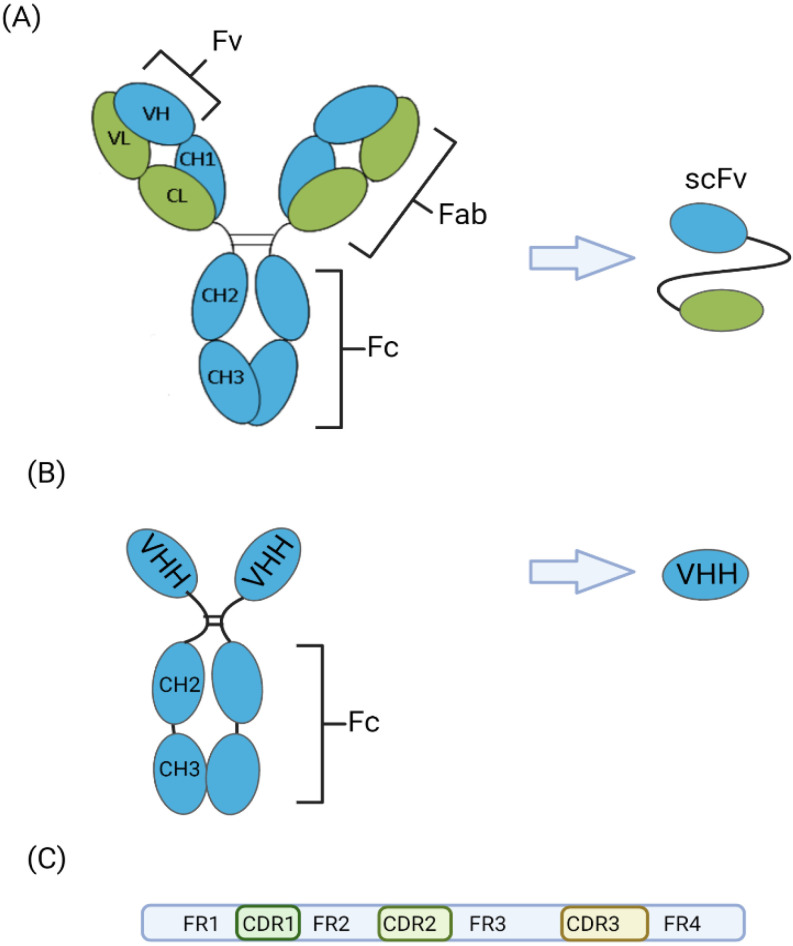
Schematics of (**A**) conventional immunoglobulin G (IgG1) showing two heavy chains (blue) and two light chains (green), which are folded into constant and variable domains. The fab domains consist of two variable domains (the variable fragment, Fv) and two constant domains acting as a structural framework. The smallest intact functional unit of IgG1 consists of a single heavy (VH) and light (VL) chain linked by an oligopeptide. (**B**) Camelid heavy-chain antibody (IgG2 and IgG3 fractions) consists of only heavy chains and the smallest functional antigen-binding fragment is the VHH, which stands for the variable domain (VH) of heavy-chain antibodies. (**C**) Each variable domain of the IgG1, IgG2 and IgG3 contains three hypervariable loops, known as complementary determining regions (CDRs) between four less variable framework regions (FR). Modified from [6]. Created with BioRender.com (accessed on 31 December 2022).

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
