# Peer review of "Therapeutic Phage Display-Derived Single-Domain Antibodies for Pandemic Preparedness"

_2073-4468, 2023, doi:10.3390/antib12010007_

Round 1

Reviewer 1 Report

A useful short commentary on potential for phage library-derived VHHs in a pandemic context.  The text gives an honest assessment of the impact of single domain antibodies as therapeutics and in diagnostic applications.  Even though variable regions are readily obtained, engineering and manufacturing are time consuming, and as we saw antibodies played a minor role compared to sequence-based vaccines with much shorter lead times and faster ability to accommodate variants. 

The one clear and differentiating advantage for VHHs (and bovine knob domains), which is mentioned, but should be made more of, is the potential for variable regions with small footprints to access relatively occluded epitopes, against which vaccines are unable to elicit conventional antibodies.  Such epitopes are often more likely to be conserved, with low variabilty, due to lack of immunity-based selective pressure.  I think this is an important point to stress in the context of the paper.  Please would the authors add a paragraph on this topic.

Author Response

Thank you for your point about the benefits of the small size of VHH, we have included this, with a reference, in a concise manner. Text added: "The relatively small size of the VHH domains also allows them to bind occluded epitopes that conventional vaccines may not be able to elicit antibodies against."

Reviewer 2 Report

With all respects to the Authors: My general comment is that this commentary to be revised completely in order to add some realistic language into the text throughout. The VHHs have many advantages and they have been rightly cited in the text; however, we should be cautious not giving the readers the hope that it could solve all the problems in relation to a future pandemic. The example of first therapeutic Nanobody almost 25 years after its discovery despite so much advancement in all molecular technologies should shed some lights to the fact that these antibody fragments have almost always been over-advertised as the magic solution and there are real issues in terms of their application, most notably the production cost. It is not cheaper to produce them in large industry scale when, for example, one needs to spend lots of efforts to remove the bacterial endotoxins for a human application. So, we need to be realistic, and I suggest the language of the text to be adjusted accordingly. 

I have added all my suggestion in the attached file.  

Author Response

Thank you for taking the time to review this commentary. We have addressed all your comments / incorporated the suggestions made in the accompanying file. However, in section 4 where the comment is made that lateral flow technology with nanobodies are not available in developing countries was not meant to mean that the technology is readily available to develop these but rather that the technology could be applied to develop lateral flow tests that could then be accessible to developing countries.

Reviewer 3 Report

Antibodies 2051084

The manuscript submitted by Daly et al reports the commentary on therapeutic phage display single-domain antibodies for the future pandemics. This manuscript is well structured overall, and it has been found to add to current research, but it needs little modification that I have outlined in the comments below.

1. Line 40 and 41: Please be consistent with the ‘naïve’ word style.

2. Figure 1, line 52: Please correct the spelling ‘heavh’ to ‘heavy’.

3. Line 58-59 and line 65: Why these sentences/words are highlighted?

4. Line 67-68: It would be great if the authors add few more references of immunized camelid libraries.

5. Line 142: Please fix ‘Hemagglutination’ spelling.

6.  It would be great if the authors touch on the current Monkeypox outbreak, does recombinant antibodies would help?

Author Response

Thank you for your comments. These have been addressed as follows:

  1. Line 40 and 41: Please be consistent with the ‘naïve’ word style. - Corrected
  2. Figure 1, line 52: Please correct the spelling ‘heavh’ to ‘heavy’. - Corrected
  3. Line 58-59 and line 65: Why these sentences/words are highlighted? - Corrected
  4. Line 67-68: It would be great if the authors add few more references of immunized camelid libraries.- We have added three more references for immunized camelid libraries for SARS-CoV-2

Güttler T, Aksu M, Dickmanns A, Stegmann KM, Gregor K, Rees R, Taxer W, Rymarenko O, Schünemann J, Dienemann C, Gunkel P, Mussil B, Krull J, Teichmann U, Groß U, Cordes VC, Dobbelstein M, Görlich D. Neutralization of SARS-CoV-2 by highly potent, hyperthermostable, and mutation-tolerant nanobodies. EMBO J. 2021 Oct 1;40(19):e107985. doi: 10.15252/embj.2021107985. Epub 2021 Aug 9. PMID: 34302370; PMCID: PMC8420576.

Valenzuela Nieto G, Jara R, Watterson D, Modhiran N, Amarilla AA, Himelreichs J, Khromykh AA, Salinas-Rebolledo C, Pinto T, Cheuquemilla Y, Margolles Y, López González Del Rey N, Miranda-Chacon Z, Cuevas A, Berking A, Deride C, González-Moraga S, Mancilla H, Maturana D, Langer A, Toledo JP, Müller A, Uberti B, Krall P, Ehrenfeld P, Blesa J, Chana-Cuevas P, Rehren G, Schwefel D, Fernandez LÁ, Rojas-Fernandez A. Potent neutralization of clinical isolates of SARS-CoV-2 D614 and G614 variants by a monomeric, sub-nanomolar affinity nanobody. Sci Rep. 2021 Feb 8;11(1):3318. doi: 10.1038/s41598-021-82833-w. PMID: 33558635; PMCID: PMC7870875.

Anderson GP, Liu JL, Esparza TJ, Voelker BT, Hofmann ER, Goldman ER. Single-Domain Antibodies for the Detection of SARS-CoV-2 Nucleocapsid Protein. Anal Chem. 2021 May 18;93(19):7283-7291. doi: 10.1021/acs.analchem.1c00677. Epub 2021 May 6. PMID: 33955213; PMCID: PMC8117401.

  1. Line 142: Please fix ‘Hemagglutination’ spelling. - Corrected
  2. It would be great if the authors touch on the current Monkeypox outbreak, does recombinant antibodies would help? – Added line 191: With the recent emergence of mpox (previously known as monkeypox) infections, there is also a possibility that novel VHH antibodies would be developed to establish rapid diagnostics and potentially neutralizing antibodies for therapy.

Round 2

Reviewer 2 Report

The following corrections are essential: 

Line 24-26:

. The 24 first nanobody-based therapy (calpacizumab) was approved in the USA in 2019 and European 25 Union in 2020 2018 for the treatment of acquired thrombotic thrombocytopenic purpura 26 (aTTP), a rare blood-clotting disorder associated with over-production of von Willebrand 27 factor.

Line 42-42:

AlternativelyIn another approach, Tsoumpeli, et al. [7] used PCR to introduce diversity into the CDR3

Line: 50-52 (Figure 1)

Camelid heavy-chain antibody 50 (IgG2 & IgG3 fractions) consists of only of heavy chains and the smallest functional antigen-binding fragment is the 51 single-domain VHH which stands for the Variable domain (VH) of the Heavy chain antibodies. (C) Each variable domain of the IgG1, IgG2 and IgG3 contains.......

Line 95: The small size of VHH also means that they can bind to epitopes less accessible to conventional antibodies;.....

THIS IS A REPEATITIVE SENTENCE TO WHAT ALREADY MENTIONED IN THE LINE 85, PLease consider removing or consolidating it to line 85.

Line 121: ..... their half-life half-lives?

Line 134:

...assays to screen for aggregation-resistant VHHs [8] and modifications to improve......

Author Response

The following corrections are essential: 

Line 24-26:

. The 24 first nanobody-based therapy (calpacizumab) was approved in the USA in 2019 and European 25 Union in 2020 2018 for the treatment of acquired thrombotic thrombocytopenic purpura 26 (aTTP), a rare blood-clotting disorder associated with over-production of von Willebrand 27 factor.

  • Done (switched EU and USA as makes sense to present in chronological order

Line 42-42:

AlternativelyIn another approach, Tsoumpeli, et al. [7] used PCR to introduce diversity into the CDR3

  • Done

Line: 50-52 (Figure 1)

Camelid heavy-chain antibody 50 (IgG2 & IgG3 fractions) consists of only of heavy chains and the smallest functional antigen-binding fragment is the 51 single-domain VHH which stands for the Variable domain (VH) of the Heavy chain antibodies. (C) Each variable domain of the IgG1, IgG2 and IgG3 contains.......

  • Done (, which stands for the variable domain (VH) of heavy chain antibodies.

Line 95: The small size of VHH also means that they can bind to epitopes less accessible to conventional antibodies;.....

THIS IS A REPEATITIVE SENTENCE TO WHAT ALREADY MENTIONED IN THE LINE 85, PLease consider removing or consolidating it to line 85.

  • The statement has been merged to line 85.

Line 121: ..... their half-life half-lives?

  • Done

Line 134:

...assays to screen for aggregation-resistant VHHs [8] and modifications to improve......

  • Done (aggregation-resistant VHH)
